# Microstructure Evolution by Thermomechanical Processing in the Fe-10Al-12V Superalloy

**Pedro A. Ferreirós** [1,2,*], **Abraham A. Becerra** [2], **Uriel A. Sterin** [2], **Martina C. Ávalos** [3], **Raúl E. Bolmaro** [3] **and Gerardo H. Rubiolo** [2]

1   School of Metallurgy and Materials, University of Birmingham, Birmingham B15 2TT, UK
2   Instituto Sabato-Comisión Nacional de Energía Atómica (CNEA), Av. Gral. Paz 1499, San Martín, Buenos Aires B1650KNA, Argentina
3   Instituto de Física Rosario, CONICET-UNR, Ocampo y Esmeralda, Rosario S2000EKF, Argentina
*   Correspondence: p.ferreiros@bham.ac.uk

**Abstract:** Nowadays, great efforts are being made to develop bcc-superalloys for medium- and high-temperature applications. However, the high brittle-to-ductile transition temperatures (BDTT) have restricted their application. Therefore, designing hot-processing routes to obtain a refined grain in these new superalloys is required. Particularly in the Fe-10Al-12V (at%) alloy, we have recently tested the BDTT shifting and, using physical models, it was indicated that a combination of $L2_1$-precipitate sizes with small grain sizes could shift the BDTT below room temperature. Here, we will present the study that allowed us to design the processing route for grain refinement in the tested superalloy. Molds of different geometry and with metallic and sand walls were used to test two different types of casting. Carbide conditioning treatments for improving the sizes and distribution were studied. The recrystallization process was explored first by hot rolling and post-annealing in stepped geometry samples with two different columnar grain orientations. Finally, we analyzed the grain microstructure obtained along a hot processing route consisting of carbide conditioning treatment, forging into a squared bar, and hot rolling up to a 2.8 mm thickness strip.

**Keywords:** ferritic alloy; vanadium carbides; hot rolling; forging; grain size; recrystallization; EBSD





## 1. Introduction

Face-centered-cubic (fcc) nickel-based superalloys are the most used materials for high-temperature applications when resistance to creep, fatigue, and environmental degradation are required [1]. Nowadays, great efforts are being made to develop novel types of superalloys using the same design concept of Ni-superalloys, comprising a ductile matrix strengthened by coherent nanometric precipitates. Within this novel group of alloys are the (body-centered-cubic) bcc-superalloys [2–4], where the main changes for having a different structure are reflected in plastic deformation mechanisms [5–7] and higher diffusivity [8]. The diversity of potential bcc-systems offers a wide range of costs, densities, and application temperatures [2]. However, the high brittle-to-ductile transition temperatures (BDTT) have restricted their application [9–11].

In previous works [12,13], we explored similar prototype alloys produced at a small scale and with high-purity materials which consist of a ferritic A2 disorder matrix strengthened by coherent $Fe_2AlV$ ($L2_1$) precipitates. In a Fe-12Al-12V alloy with 22% $L2_1$ volume fraction, the strengthening peak was obtained for a 10 nm precipitate radius. The alloy showed a low lattice misfit of −0.014% [14] which could be increased by Ti addition [15]. Charpy samples of the alloy, with a microstructure of equiaxed grains averaging 700 μm in size, were used to assess its BDTT value [16]. A high BDTT value was obtained, around 617 °C, but well below the BDTT values of other bcc superalloys. Our discussion of results in the last reference paper suggested that by careful control of grain size and precipitate's coherency, through its size, an alloy may be made that is both resistant to deformation and tough.

In this context, grain size control can be attempted through conventional thermomechanical routes that can produce recrystallization and grain refinement, such as hot extrusion, rolling, or forging. However, in such a case, larger ingots of the prototype alloy must be cast and therefore high purity materials cannot be used. It should also be noted that this type of superalloy has high strength and low ductility at RT (dual-phase field), eliminating the possibility of cold working and static recrystallization (SRX) by annealing treatments.

In polycrystal alloys, a common method to reduce grain size is through the mechanism of recrystallization [17]. However, the difficulty to initiate a recrystallization mechanism in ferritic-based alloys by thermomechanical processes is well known [18]. Instead, dynamic recovery (DRV) is easily initiated in ferritic alloys during metalworking. The recovery is defined as all annealing processes occurring in deformed materials where the migration of high-angle grain boundaries is absent [19]. The trend of ferritic structures to promote the dislocation motion during deformation and inhibit the grain nucleation is due to its high stacking fault energy [20]. Nevertheless, a dynamic recrystallization process (DRX) of these alloys is possible under high strains and strain rates [21,22].

In this work, we report the studies that allowed us to obtain a thermomechanical process for a 50 mm diameter cylindrical ingot (2 kg) of the Fe-10Al-12V (at. %) alloy, produced from the casting of low purity materials, until obtaining a 2.8 mm thickness strip with small grain size. This material was recently used to study the synergy between the size of the grain and the precipitate to change the value of its BDTT [23]. The results of the high-temperature Charpy test showed a substantial BDTT decrease with grain size refinement and precipitate coarsening. The minimum BDTT obtained was 341 °C for the alloy with a grain size average of 38 μm and precipitate size of 61 nm and the yield stresses were above 600 MPa for temperatures below 500 °C. While the BDTT of the ferritic A2 matrix was 154 °C for a grain size average of 110 μm and yield stresses above 350 MPa for temperatures below 500 °C. Moreover, these experimental results were modeled in such way that we can predict the nil ductility temperature (NDT) of this alloy as a function of grain and precipitate size, the prediction allows us to establish NDT values below room temperature if the grain and precipitate sizes vary between 20 to 10 μm and 70 to 65 nm respectively. As it will be demonstrated by the results reported in this work, the range of the alloy grain size to reach the objective of reducing the NDT to technologically competitive values can be achieved with the thermomechanical process described below.

The studies included several stages. As a starting point, a prismatic ingot was cast using a mold with steel walls. Several isothermal treatments on samples of this prismatic ingot, with different times and cooling rates, were performed to reduce and homogenize the distribution of the as-cast carbides. Knowing the best treatment for carbides, it was applied to the rest of the ingot and then, from perpendicular cuts of the ingot, two stepped samples with different preferential orientations of columnar grain were obtained. Next, hot rolling (in A2 single-phase field) and post-annealing of the two staggered samples were carried out. The grain distribution and restoration mechanisms during hot rolling, and post-annealed, were analyzed with electron backscatter diffraction (EBSD) techniques. The results obtained up to this stage of the studies allowed us to propose a processing route consisting of casting in a cylindrical sand mold, carbide conditioning treatment, square bar forging and hot rolling up to a 2.8 mm strip. This study, initially aimed at reducing the BDTT of ferritic superalloys, can also be used as an optimal processing window for hot forging of a Fe-Al-V ferritic superalloy that will produce defect-free parts during industrial-scale manufacturing.

## 2. Materials and Methods

### 2.1. Alloy Production and Processing

The alloys were produced by air induction melting equipped with a gas-burning system on the surface to minimize oxidation. Raw materials of mild steel SAE 1005 (0.05% C), aluminum sheets (99.9%), and ferrovanadium lump (80% V, 20% Fe, 0.09% C) (wt.%) were used. Two types of molds were explored: (a) Prismatic ingot with steel walls and (b) cylindrical ingot with sand walls.

### 2.1.1. Carbide Conditioning Treatments Exploration

Samples from the prismatic ingot were used to explore thermal treatments for conditioning the size and distribution of carbides. The homogenization heat treatments were conducted in an electric furnace under an argon atmosphere at 1100 °C for 1 h, 4 h, and 16 h. Water and air cooling (natural convection) media were explored after the 4 h treatment.

### 2.1.2. Prismatic Ingot Processing

The alloy was cast into a prismatic steel mold of 80 × 60 × 30 mm (20 mm of wall thickness). A truncated inverted rectangle pyramid riser with 62% of the volume mold was used. A fully columnar solidification structure was obtained due to the rapid heat extraction rate towards the metallic walls of the mold. The ingot sectioning was performed by electric discharge machining (EDM). Figure 1a shows the scheme of ingot sectioning to obtain samples for as-cast metallography, carbide conditioning treatments, and hot rolling.

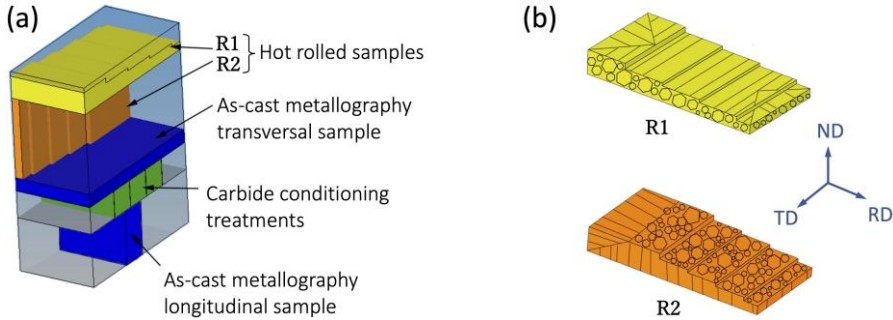

**Figure 1.** Prismatic ingot. (**a**) Sectioning scheme. (**b**) CGPO scheme profile in samples R1 and R2. The lines and hexagons represent the growth direction and the cross-section of columnar grains, respectively. The directions (D) used for hot rolling are indicated as N (normal), R (radial) and T (transversal).

Two stepped samples, R1 and R2, with different columnar grain preferential orientation (CGPO) were obtained from perpendicular cuts of the ingot (Figure 1a). A carbide conditioning treatment at 1100 °C for 4 h with subsequent air cooling was applied in both samples. The stepped geometry of R1 and R2 samples (variable heights from 8.75 to 2.5 mm) allows several strains ($\varepsilon$) and strain rates ($\gamma$) to be applied in a single rolling pass (see Table 1). The samples were heated at 900 °C under an argon atmosphere and hot-rolled in a 50 HP Krupp machine with a 210 mm roll diameter at 54 rpm up to 3.2 mm, leaving the smallest step without deformation as a reference microstructure. The CGPO of the sample is aligned with the transverse rolling direction (TD) for the R1 and with the normal rolling direction (ND) for R2 (Figure 1b). The stepped geometry was designed to explore a strain rate between 1 s$^{-1}$ and 30 s$^{-1}$, because between these strain rates was observed the transition from continuous dynamic recovery to discontinuous dynamic recrystallization by compression in the alloy Fe-12Al-12V (%at.) ([24], see Supplementary Materials). Finally, a post-rolling annealing treatment at 850 °C for 0.5 h was carried out to study the grain size evolution (microstructure comparison in the as-rolled and the post-annealing states).

**Table 1.** Strains and strain rates applied to the stepped R1 and R2 samples in the hot rolling process.

| Step | Initial Thickness (mm) | Rolled Thickness (mm) | $\varepsilon$ | $\gamma$ (s-1) |
|---|---|---|---|---|
| 1 | 2.50 | 2.50 | 0 | 0 |
| 2 | 4.54 | 3.20 | 0.35 | 20.24 |
| 3 | 5.65 | 3.20 | 0.57 | 24.31 |
| 4 | 7.08 | 3.20 | 0.80 | 26.98 |
| 5 | 8.75 | 3.20 | 1.00 | 28.57 |

2.1.3. Cylindrical Ingot Processing

A total of 2 kg of the alloy was cast into a cylindrical sand mold of 50 mm diameter and 80 mm height. A conical open riser with 50% of the cylindrical ingot volume was used. The ingot was sectioned by EDM at 40 mm of height and the ingot base half was used for the subsequent processing. A carbide conditioning treatment of 4 h at 1100 °C with air cooling was applied (see estimated cooling rate in Supplementary Materials, [25–28]). The ingot was forged up to a square cross-section of 20 mm of sides in an open die with a 100 kg pneumatic hammer, Bêché & Grohs L3 (200 hits/min and 1.62 kJ impact energy), at temperatures between 1100 °C and 900 °C (A2 single-phase field [29]). Using the means cross-section (*S*) of the ingot ($S_0$) and the forged bar ($S_f$), a coefficient of deformation of $\lambda = \left( S_0/S_f \right)$ = 4.9 was obtained. A length of 82 mm of the bar was cut, heated in Ar flow at 970 °C, and hot-rolled at 54 rpm in a 50 HP Krupp with Ø 210 mm rolls. The bars were rolled up to 2.8 mm thickness in five steps with an incremental true strain of $\varepsilon$ = 26, 33, 40, 47 and 52%.

*2.2. Characterization*

Phase transformation measurements were made by DSC in Setaram Labsys Evo equipment using an alumina crucible, dynamic argon atmosphere (25 mL/min), and heating rates of 5 °C/min. A sample of 3 mm diameter and 100 mg from the prismatic ingot was used for the DSC measurement. The transformation temperatures correspond to the extrapolated transition onset that appears in DSC signals during heating.

Specimens for microscopy observations were ground up to 600 grit, then polished with 1 μm diamond paste and finally etched in an acidic solution (68% glycerin + 18% HNO3 + 18% HF). As-cast macrostructures were observed by a stereo microscope Olympus SZ-ST. Microstructures were inspected by an Olympus BX60M optical microscope and a Phillips QUANTA 200 scanning electron microscope (SEM) equipped with an EDAX energy dispersive X-ray spectrometer (EDS). The main alloying elements were quantified by wavelength dispersive X-ray fluorescence (WDXRF) in Bruker S8-Tiger equipment (see Table 2). The C content was measured by atomic absorption spectroscopy (AAS) according to ASTM E1019. EBSD scans were performed on transverse sections of R2 sample with $\varepsilon$ = 1.00 in FEI FEG-SEM Quanta 200 microscope and TSL-OIM system, DigiView camera and TSL-OIM 7.3 post-processing software. Samples were polished by different grit sandpapers, followed by 9, 6, 3 and 1 μm diamond pastes and vibratory colloidal silica for 30 min. A step size of 2.0 μm was chosen due to the large areas of scan and the large grain sizes observed.

**Table 2.** Chemical compositions by WDXRF except for C by AAS (at. %). The error represents the standard deviation obtained from different zones of the samples.

| Ingot | Fe | Al | V | Si | Mn | C |
|---|---|---|---|---|---|---|
| Prismatic | 77.1 ± 0.3 | 10.2 ± 0.2 | 12.1 ± 0.1 | 0.38 ± 0.01 | 0.30 ± 0.01 | 0.26 ± 0.03 |
| Cylindrical | 77.0 ± 0.8 | 10.0 ± 0.5 | 12.2 ± 0.5 | 0.42 ± 0.01 | 0.39 ± 0.01 | 0.27 ± 0.05 |

## 3. Results and Discussion

The casting method has led to a loss of about 2 at.% Al with respect to the raw materials for both types of ingots. The aluminum was probably depleted due to the formation of $Al_2O_3$, which rises to the surface of the molten metal as a slag [30,31]. The compositions obtained for both ingots reflect similar results, as shown in Table 2.

*3.1. As-Cast Microstructure of Prismatic Ingot*

The as-cast macrostructure showed a complete columnar growth in two central sections of the ingot (Figure 2). The mold geometry allowed an equal heat removal during solidification, producing an angle of about 45° between columnar fronts (Figure 2). The as-cast microstructure characterization showed the presence of carbides located inside of

the grains and along the grain boundaries. EDS measurements indicated high contents of C and V. The carbides had a plate-like morphology (faceted) with an average thickness of 1.1 ± 0.5 μm (Figure 3a) in most of the ingot section; however, in the core region, an irregular carbide morphology (non-faceted) of 2.2 ± 0.5 μm (Figure 3b) was observed. Similar carbide morphologies were identified in $Fe_3Al$ alloys with different C contents [32], while in Fe-Al-V alloys with low C content (~200 ppm) only faceted vanadium carbides were observed [12]. The change of carbide morphology in the core ingot zone could be due to C macro-segregation, which is a common feature in ferritic alloys [33].

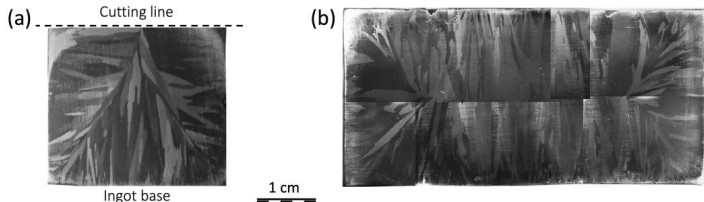

**Figure 2.** As-cast macrostructures of the prismatic ingot. (**a**) longitudinal section, (**b**) transversal section (details of the ingot sectioning in Figure 1a).

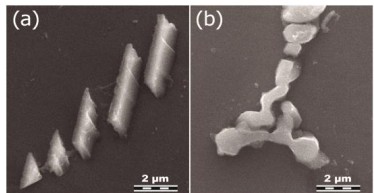

**Figure 3.** SEM micrographs of the as-cast Fe-10Al-12Valloy showing the morphologies of carbides: (**a**) plate-like (faceted) and (**b**) irregular (non-faceted).

### *3.2. Phase Transitions*

The DSC curves measured for Fe-12Al-12V [15] (using high-purity raw materials) and Fe-10Al-12V (low-purity raw materials) are shown in Figure 4. The Fe-10Al-12V phase transition temperatures measured on heating (Figure 4a), presented the transformations $A2 + L2_1 \rightarrow B2 + L2_1$ and $B2 \rightarrow A2$ at 724 °C and 785 °C, respectively. Both transitions were shifted to lower temperatures with respect to those observed in the high-purity Fe-12Al-12V alloy [15]. Although the content of impurities can change the transition temperatures, we suppose that the main cause is due to the reduction of Al (melting process) and V (formation of vanadium carbides). The observed differences in the transformation temperatures with respect to the Fe-12Al-12V alloy [15] are consistent with the composition deviations according to trends indicated by the Fe Al V phase diagram [12].

The water quenching with at least 4 h of treatment generates small-sized and homogeneously distributed vanadium carbides in grain and grain boundaries (Figure 6b,c). However, extremely high cooling rates in combination with the brittleness of these alloys could promote the formation of microcracks, especially in pieces of complex geometries. Therefore, we also tested an isothermal treatment for 4 h at 1100 °C followed by air cooling (natural convection). This sample showed no substantial differences to those observed in water quenching by SEM (Figure 6b). In view of these results and for the following hot processing steps of the alloy, we selected for the carbide conditioning the 4 h isothermal treatment at 1100 °C with a post air cooling. The estimated cooling rate is presented in the Supplementary Materials.

The intermediate temperature of phase transition $B2 + L2_1 \rightarrow B2$ was not detected by DSC heating. On the other hand, this phase transition is clearly detected on DSC cooling but has an undercooling temperature shifting, as it was previously found in Fe-Al-V and Fe-Al-V-Ti alloys [15]. Figure 4b shows the $B2 \rightarrow B2 + L2_1$ transition at 759 °C for the Fe-10Al-12Valloy. In this case, the phase transition was at a higher temperature than the Fe-12Al-12V alloy in cooling but is still under the 764 °C of $B2 \rightarrow B2 + L2_1$ detected in DSC heating for the Fe-12Al-12V alloy.

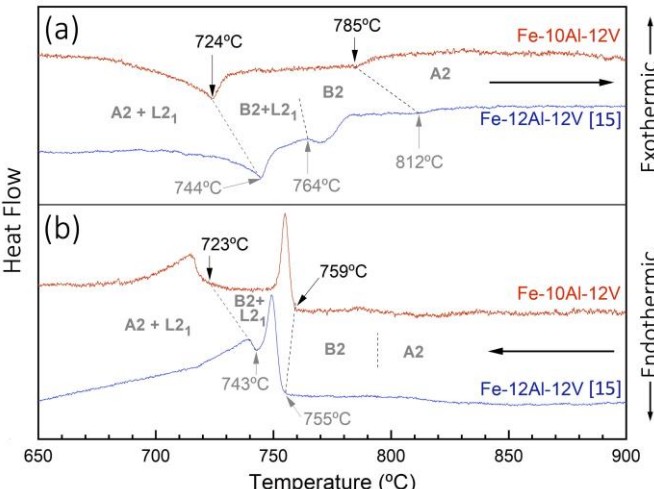

**Figure 4.** DSC in Fe-10Al-12V and Fe-12Al-12V [15] alloys. Heating (**a**) and cooling (**b**) at 5 °C/min.

### 3.3. Carbide Conditioning Treatments

The carbide dispersion and their size have a strong effect on the behavior of the material during hot working [34] as well as in finished products [35,36]. In general terms, continuous precipitation of carbides along the grain boundaries can lead to the embrittlement of the material. Hence, thermal treatments were explored in order to homogenize the carbide dispersion, reduce their size and impede their continuity on grain boundaries. Figure 5 shows the dispersion of vanadium carbides in the as-cast condition and after 1 h, 4 h and 16 h treatments at 1100 °C followed by water quenching. These observations show that the carbides reduce their size and lose their continuity along the grain boundaries as the treatment time increases. Only a few large carbides (micrometric sizes) were found after 4 h of treatment and none were observed after 16 h. With a higher magnification (Figure 6), it is possible to find a high amount of nanometric carbides (160 ± 27 nm) after 4 h of treatment. It should be noted that after this treatment time, the carbides that precipitated in grains and in grain boundaries have similar sizes. These results suggest that most micrometric carbides from the solidification process can be dissolved in the ferritic matrix after 4 h of treatment at 1100 °C.

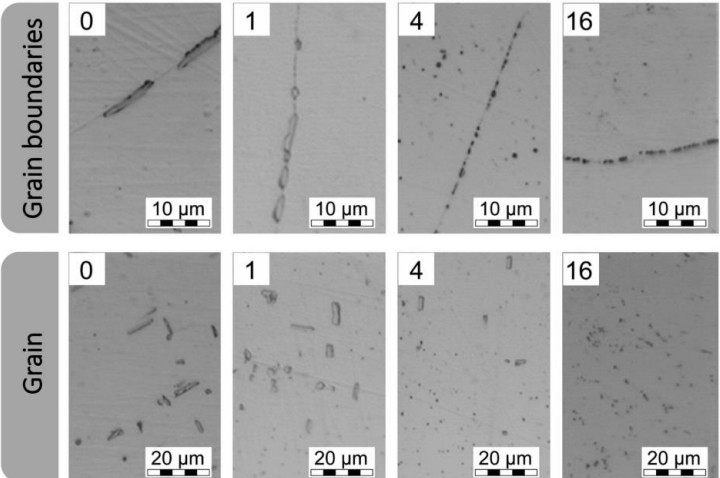

**Figure 5.** Optical micrographs of vanadium carbides evolution after isothermal treatments at 1100 °C for different times and with water quenching in samples from the prismatic ingot. The treatment times are 0 (as-cast), 1, 4, and 16 h (time indicated in the upper left corners of each micrograph).

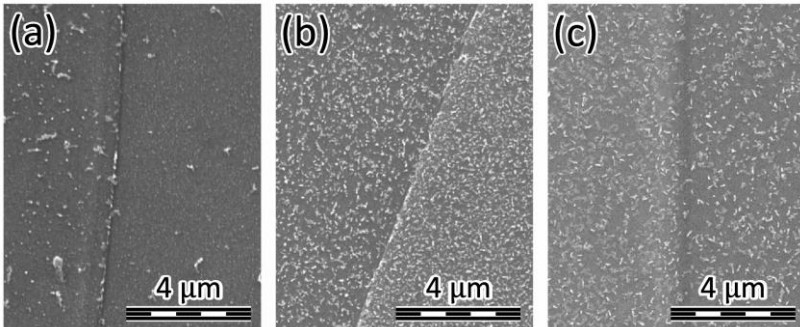

**Figure 6.** SEM micrographs showing the nanometric carbides formed in grain boundary zones after 1100 °C isothermal treatments with water quenching at: (**a**) 1 h, (**b**) 4 h and (**c**) 16 h.

### 3.4. Hot Rolling and Annealing of Stepped Samples

Two stepped samples (R1 and R2) with perpendicular CGPO (Figure 1b) were hot rolled at 900 °C in a single pass. Figure 7 shows the microstructure along different stepped sections of the rolled samples. The columnar grains of R1 (CGPO//TD) suffered ovalization of their cross sections along the RD direction, while R2 (CGPO//ND) showed folded grains in the ND direction. The folded grains curvature reveals the strain distribution along the thickness of the strip, which is typical of the rolling process as shown in [37] using a pin embedded in the strip. In both samples, for $\varepsilon \geq 0.57$, two zones are clearly distinguishable: (a) surface zones (equiaxed grains) and (b) central zone (deformed grains), which suggest that DRX and DRV were, respectively, activated in these regions. The two zones for each section were delimited schematically by dashed lines in Figure 7. The equiaxed grains reached an area of $40 \pm 6\%$ (top + bottom surfaces) for $\varepsilon = 1.00$. The perpendicularity between the CGPO of R1 and R2 did not produce a distinguishable effect on the grain refinement fraction. Therefore, the following results and discussion are presented only for the R2 sample. Figure 8 shows optical micrographs of the surface and the central zones. Small recrystallized equiaxed grains can be observed on the rolled surface at $\varepsilon \geq 0.57$ (Figure 8a). On the contrary, sub-grain boundaries are observed at lower strain ($\varepsilon = 0.35$) of the surface zone (Figure 8a) and for all cases of the central zone (Figure 8b). These sub-structures could be attributed to a DRV mechanism. It should be noted that not all high-angle misorientation boundaries were clearly revealed on the optical micrographs, and although these were used here to suggest the mechanisms operating in each zone, these results will be complemented and discussed later by EBSD techniques.

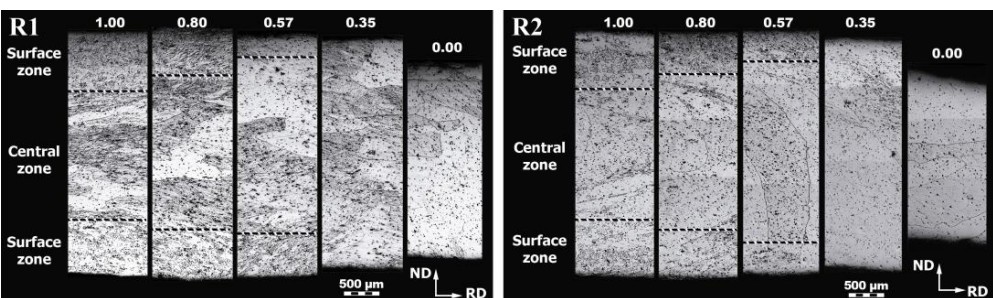

**Figure 7.** Optical macrographs for each rolled step in the transversal section of R1 and R2 samples. The numbers indicate the $\varepsilon$ values. Dotted lines delimit the surface and central zones where the grain changes.

It is possible to observe a high content of dark particles of about 10 μm in Figure 8. These particles showed a high content of Al by SEM-EDS. Despite the fact that we do not quantify their composition, we assume that they are alumina particles ($Al_2O_3$) from the melting process. In steels, it is known that the main problem of $Al_2O_3$ inclusions is the generation of surface defects by cold rolling [38,39], but in the present alloy, the brittleness given by the presence of $L2_1$ precipitates does not allow cold forming processing.

Additionally, $Al_2O_3$ particles can affect the mechanical properties of the alloy, therefore, their control in the casting process should be studied in the future [40–42].

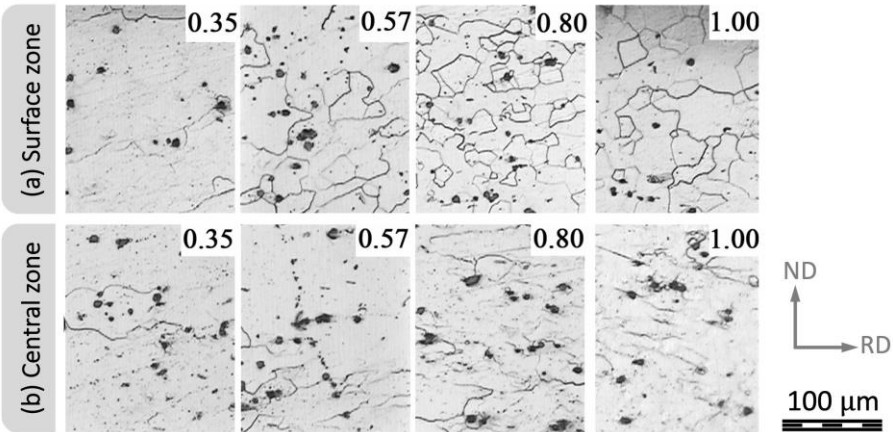

**Figure 8.** Optical micrographs along the transversal section for each deformed step in the R2 sample. (**a**) Surface zone, (**b**) central zone. The numbers indicate the $\varepsilon$ values.

### 3.5. Grain Distribution and Restoration Mechanisms

Due to the low percentage of recrystallized areas achieved by hot rolling in the stepped geometry samples, only the recrystallization for the maximal deformation ($\varepsilon = 1.00$) will be analyzed below for being more relevant to the work aim. The columnar grain sizes in the as-cast prismatic ingot were around 550 µm in diameter and a few millimeters in length. The EBSD of Figure 9a shows the grain sizes obtained for $\varepsilon = 1.00$. A grain area size average of $600 \pm 150$ µm$^2$ in the surface zones was observed (Figure 9a). These grain areas can be idealized as circular grains of average diameter $\bar{d} = 28 \pm 3$ µm. On the other hand, the central zone shows large deformed columnar grains that came from casting, although some isolated clusters of recrystallized grains can be observed as well. The inset of Figure 9a shows that many grains in the surface regions have gone through DRX during hot rolling; meanwhile, there are still some of them with a high internal crystal misorientation that will further contribute to recrystallization during post-annealing.

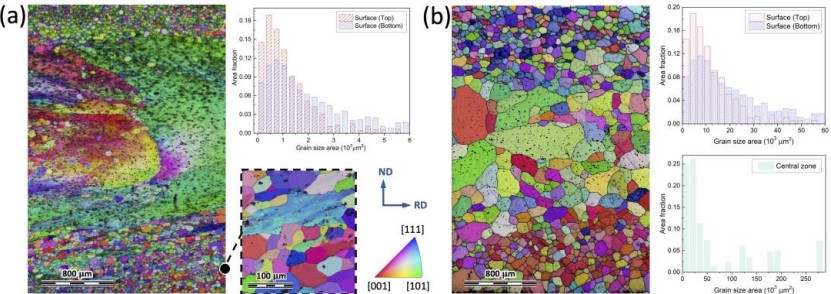

**Figure 9.** EBSD map and grain size distribution for the transversal section of R2 sample with $\varepsilon = 1.00$. (**a**) As-rolled sample. (**b**) Sample with post-rolling annealing at 850 °C for 0.5 h.

A post-annealing of 850 °C for 0.5 h was applied to the rolled R2 sample to activate an SRX process in the central zone of the rolled material. Figure 9b shows the EBSD map after the annealing. The first difference, with respect to the as-rolled sample, is the higher number of recrystallized grains in the central zone. This fact is evidence that, despite the high tendency of recovering of the ferrite [43], the deformation energy stored in the central zone by hot rolling was enough to activate an SRX process in the post-annealing. Figure 9b also shows a heterogeneous grain size in the central zone of the sample; the left side has some larger grains ($\bar{d} \approx 600$ µm) with orientations possibly related to the as-cast columnar grains but the right side showed smaller grain sizes ($\bar{d} = 172$ µm) with random orientations.

The second difference observed after annealing was the grain growth in surface zones. For these regions (top + bottom), the grain size distribution peaks showed the same value, $\bar{d}$ = 80 μm. This represents about 300% of grain growth. The grain size kinetics during grain growth has been extensively studied and can be generally described by lognormal, Weibull, or Rayleigh distribution functions [44–48]. On the other hand, the description of the grain distribution kinetics in a DRX is more complex due to the nucleation of new grains and often computational simulations are needed [49]. The cumulative distribution function for the Weibull distribution is as follows:

$$F(d) = 1 - \exp\left[-\left(\frac{d}{\alpha}\right)^{\beta}\right], \ \alpha = \frac{\langle d \rangle}{\Gamma(1 + 1/\beta)} \tag{1}$$

where $d$ is the grain size, $\langle d \rangle$ is the median grain size, $\beta$ is the shape parameter, $\alpha$ is a time-dependent parameter and $\Gamma$ is the gamma function. The experimental grain size from the surface zones in the as-rolled and annealed R2 sample were fitted by Equation (1) (Figure 10a). An initial difference is observed between the top and bottom zones in the as-rolled sample, possibly linked to an uneven deformation by the non-symmetric sample geometry. Figure 10a shows how the $\beta$ values are increasing by the grain growth during annealing. Figure 10b compares the four cumulative distribution functions of the surface zones. The as-rolled top zone is the one that moves the farthest from the group and it has a distribution closer to a Rayleigh distribution function (Rayleigh distribution function is a special case of the Weibull distribution function where $\beta$ = 2). This is also the one with the smallest grain size of the four analyzed areas; therefore, this low $\beta$ value could be due to being closer to the end of the complete recrystallization (nucleation and growth process).

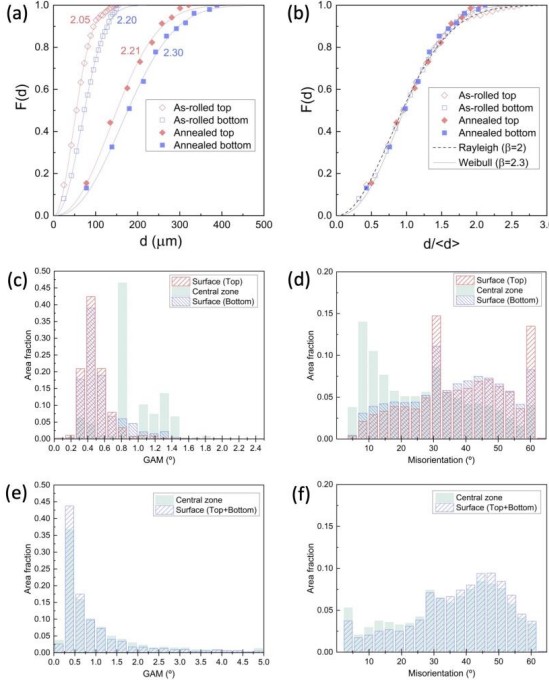

**Figure 10.** Sample R2 with $\varepsilon$ = 1.00. (**a**) Grain size cumulative distribution function in as-rolled and annealed samples. Fitted values by Weibull function ($\beta$ value is indicated for each curve). (**b**) Comparison of cumulative distribution functions. Grain average misorientation (GAM) and Misorientation histograms in (**c**,**d**) as-rolled and (**e**,**f**) annealed conditions.

The recrystallization process can be analyzed deeper observing the grain average misorientation (GAM) and misorientation distributions [50,51]. Since GAM is a measure of the internal crystallographic misorientation within grains its value represents the energy stored due to internal defects. The GAM is used to determine the recrystallized and non-

recrystallized fractions. Large GAM values can be associated with dislocation structures of recovered or partially recrystallized regions, while low GAM values indicate a low dislocation density typical of recrystallized zones. Some authors analyze these fractions by splitting the GAM into a bimodal distribution, for example, some authors used a fixed cutoff value (0.6° [52], 1° [51], 1.55° [53]) and others a value that even changes with the annealing time (between 0.8° and 1.2° [54]). The GAM histogram of the as-rolled sample, Figure 10c, shows a distribution with a principal peak maximum at 0.45° indicating that nearly a full DRX was achieved in the surface zones. Particularly for the case of the bottom surface, a small remnant of area fraction is seen in the tail of the distribution at higher GAM values, which coincides with the observations of some isolated grains with high internal crystal misorientation, as shown in the inset of Figure 9a. On the contrary, the center zone shows a more complex GAM distribution (bimodal or maybe tri-modal) with preponderance in angles greater than 0.75°, which indicates a significant fraction of non-recrystallized grains. These results are consistent with the optical observations (Figure 8).

After the annealed treatment, the GAM distribution for the surface and center zones are practically the same (see Figure 10e). These GAM distributions showed a narrow single peak at 0.37°, barely in the limit of detection of the technique, which indicates a low dislocation density and a fully recrystallized microstructure. Moreover, the histograms of misorientation angles in Figure 10e,f support these results. Generally, low angle grain boundaries (<15°) imply higher migration rates [55] which are representative of a grain substructure. Only for the central zone in the as-rolled sample, a majority of low misorientation angles were observed (Figure 10d). Instead, the surfaces showed higher misorientation angles. Furthermore, the annealed sample (Figure 10f) showed a more homogeneous distributed histogram without significant differences between the surfaces and central zones, indicating that the whole sample thickness achieved complete recrystallization.

A further understanding of the heterogeneities observed between surfaces and central regions of the samples can be obtained through the analysis of textures. Textures were determined by EBSD taking advantage of the very large areas scanned in each sample, before and after annealing. Figure 11a and 11b show the texture of the as-rolled and annealed samples, respectively. They show usual pole figures for rolled BCC materials at different temperatures and after annealing. However, there are regions where shearing strains have been exerted the most, top and bottom regions, contributing differently to the as-rolled macroscopic texture, as shown in Figure 12, where the pole figures are presented from the TD direction, for making the shearing effect more evident. The redundant shearing strains, shown by the arrows on the pole figures, make the DRX more active on the surface regions because of larger accumulated deformation energy. The heterogeneity developed mainly due to the redundant extra shearing deformation exerted by the rolls during deformation at high temperature.

In summary, the effective strain near the surface is higher than the nominal strain calculated from the reduction in thickness after hot rolling, as shown in steels by previous authors [37]. The maximum effective strain increases with increasing reduction. A band of fine recrystallized grains appears in the severely sheared region while recrystallized grains do not nucleate at the central part of the thickness of the strip. In conclusion, the results indicate dynamic recrystallization at a temperature of 900 °C if the strain rate is much greater than 29 s$^{-1}$, the maximum nominal strain rate tested. It is not possible to specify how much higher the strain rate should be, but it follows that forging rather than rolling is the appropriate method.

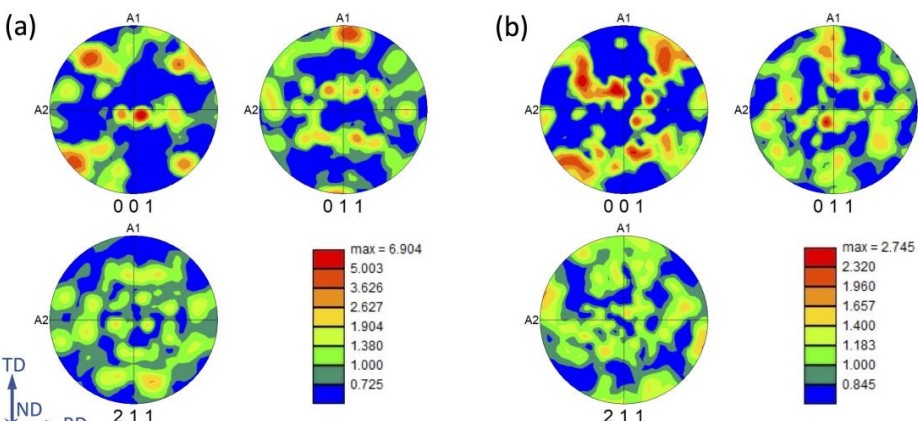

**Figure 11.** Pole figures projected on ND sample planes of R2 sample with ε = 1.00. (**a**) As-rolled sample. (**b**) Sample with post-rolling annealing at 850 °C for 0.5 h.

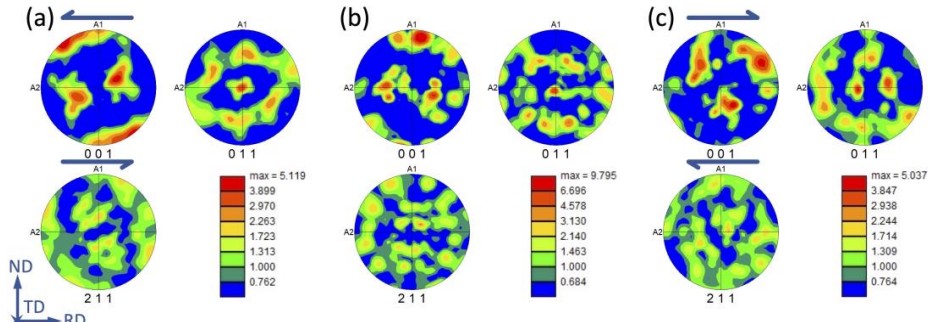

**Figure 12.** TD pole figures in the as-rolled R2 sample with ε = 1.00 for regions: (**a**) Top, (**b**) Central, and (**c**) Bottom. The arrow indicates redundant shearing strain.

### 3.6. Thermomechanical Processing in the Cylindrical Ingot

Although the processing of the prismatic ingot did not achieve a fine grain size for the full thickness of the final strip, the results allowed us to design a more appropriate thermomechanical processing route. The scheme of this hot processing route, starting from a cylindrical ingot, is shown in Figure 13a. The last step of the processing route is the aging treatment. The treatment produces a precipitation strengthening by the coherent nanometric $Fe_2AlV$ ($L2_1$) phase. The coarsening rate and possibilities of aging were extensively studied in [13,15,56], therefore they are not discussed in the present work.

Photographs of the samples with their respective dimensions used throughout the processing route are shown in Figure 13b,e. The grain microstructures obtained in each stage of processing are shown in Figure 13f,h. The cross-section of the cast ingot shows equiaxial shape grains of several millimeters in the core and radially columnar grains in the rest. The cross-section of the forged bar, Figure 13g, has grain sizes between 70 μm and 1 mm. The core and, in second place, the diagonals of the square section show the smallest grain regions due to the inhomogeneous strain distribution in the open die process [57,58]. After rolling (Figure 13h) is observed a similar effect to the previous section, where a smaller grain size (22 μm) is obtained on the surface and larger in the central zone (51 μm) of the strip. Since forging and rolling produce opposite grain refinement effects between the center and the periphery, it is possible to optimize the thermomechanical process to improve the homogeneity of grain size throughout the strip thickness. For example, achieving a homogeneous grain size of 22 μm allows obtaining an NDT of 160 °C for a precipitate radius of about 60 nm [23]. These can even be reduced to values below room temperature with larger precipitate sizes (>70 nm) or with grain sizes below 10 μm.

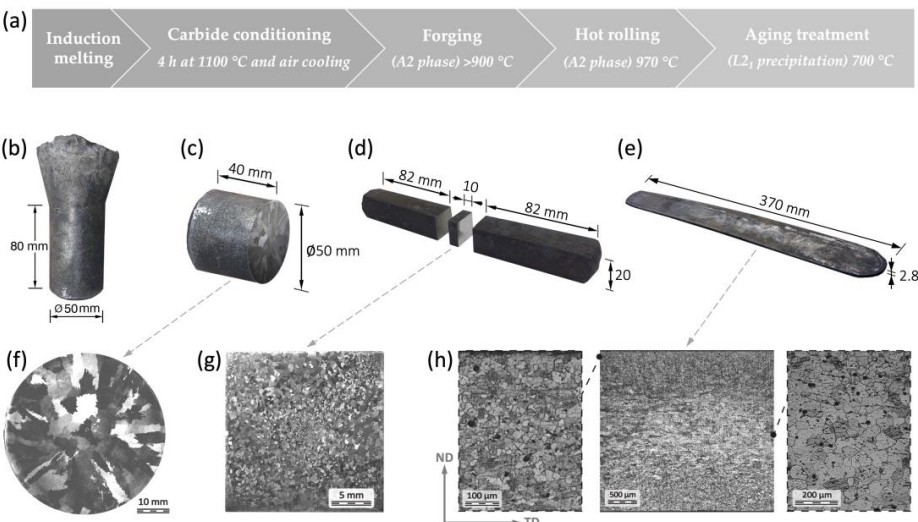

**Figure 13.** Thermomechanical processing route, samples with dimensions, and grain microstructures throughout the processing. (**a**) Schematic processing route. (**b**) Melted ingot (including riser). (**c**) Section of the ingot for forging. (**d**) Forged bar with cuts for rolling and microstructural observations. (**e**) Hot rolled bar. (**f**) Cross-sectional macrograph of the cast ingot. (**g**) Cross-sectional micrograph of the forged bar. (**h**) Cross-sectional micrograph of the hot rolled bar including surface and central zones enlargements.

## 4. Conclusions

This study provides first-time insights into a thermomechanical processing route for strip production of a ferritic $Fe_2AlV$-strengthened alloy. The ternary Fe-10Al-12V (at.%) alloy with high content of impurities was explored.

The thermomechanical processing route consists of several stages. First, raw materials of mild steel, aluminum sheets and ferro-vanadium lump are melted in an air induction furnace equipped with a gas-burning system on the surface to minimize oxidation. Then, the melt is cast into a cylindrical sand mold. This is followed by a carbide conditioning treatment of the cylindrical ingot by heating it in an electric furnace at 1100 °C for 4 h under an argon atmosphere and air cooling. Later, begins the hot open die forging process at 900 °C to convert the cylindrical ingot to a bar until a forging coefficient of deformation of $\lambda = 4.9$ is reached. The semi-finished square-shaped bar showed grain sizes between 70 μm and 1 mm. The diagonals of the square section show the smallest grains due to the inhomogeneous strain distribution in the open die process. A subsequent hot rolling at 900 °C was used to homogenize and refine the grain size, obtaining average sizes of 22 μm and 51 μm in the surface and central zones of the strip, respectively. Higher strain during hot rolling will increase the inhomogeneous strain distribution, decreasing the grain size further below the surface of the strip than at half the thickness. Therefore, if a smaller and more homogeneous grain size is wanted, it is necessary to utilize a higher forging coefficient of deformation.

**Supplementary Materials:** The following supporting information can be downloaded at: https://www.mdpi.com/article/10.3390/alloys2010002/s1, Figure S1: Estimation of air-cooling rates (natural convection) of samples; Figure S2: Uniaxial compressive test at 900 °C for different strain rates in the Fe-12Al-12V (%at.) alloy. Adapted from Ref. [24].

**Author Contributions:** Conceptualization, P.A.F., A.A.B., U.A.S. and G.H.R.; formal analysis, P.A.F.; investigation, P.A.F., A.A.B., U.A.S., M.C.Á. and R.E.B.; resources, R.E.B. and G.H.R.; data curation, P.A.F., A.A.B., U.A.S. and G.H.R.; writing—original draft preparation, P.A.F., A.A.B. and U.A.S.; writing—review and editing, P.A.F., M.C.Á., R.E.B. and G.H.R.; visualization, P.A.F., A.A.B. and U.A.S.; supervision, G.H.R. All authors have read and agreed to the published version of the manuscript.

**Funding:** P.A. Ferreirós gratefully acknowledge funding from UKRI (Grant No. MR/T019174/1).

**Data Availability Statement:** Not applicable.

**Acknowledgments:** A.A. Becerra and U.A. Sterin were supported by OEA and CONICET scholarships, respectively. The authors would like to thank P. Bozzano and R. Castillo Guerra, heads of Electron Microscopy and Metallography CNEA's laboratories, respectively, for their collaboration and availability of equipment. We acknowledge the collaboration of Vanina Taralini and Pablo Risso for the finely performed EBSD experiments.

**Conflicts of Interest:** The authors declare no conflict of interest.

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
