# Peer review of "Microstructure Evolution by Thermomechanical Processing in the Fe-10Al-12V Superalloy"

_alloys, doi:10.3390/alloys2010002_

Round 1
Reviewer 1 Report
The author presented " Microstructure evolution by thermomechanical processing in the Fe-10Al-12V superalloy" the authors used , many tools (EBSD, OM, EDS etc.,) and also presented an depth analysis. Manuscript is well written and novel up to some extent. It can be published but after some minor concerns.
"
1. abstract and introduction both are well written, however a little more emphasize on the problem statement together with your prior publications can increase the the soundness of the introduction.2. Fig. 1, please check the font size of labels, it should be same, similar Fig. 4, 5, 6
3. Fig.2 is not visible , increase the size of the image
4. otherwise it is a well established study.
Author Response
Thank you very much for your comments. Please find attached the new version of the manuscript with the corrections marked in red.
1) The introduction was modified to achieve the reviewer's suggestion (paragraphs starting on line 44 and line 74).
2) The font sizes in Fig. 1 were unified. With the same aim, the following figures were modified: 2, 3, 4, 5, 6, 8 and 13.
3) The size and quality of Figure 2 were increased.

Reviewer 2 Report
This paper aims to investigate the microstructure evolution by thermomechanical processing in the Fe-10Al-12V superalloy. A processing route was designed for grain refinement in the tested superalloy. The recrystallization process was explored first by hot rolling and post-annealing in stepped geometry samples with two different columnar grain orientations. Finally, the grain microstructure obtained along a hot processing route was analyzed. Conclusions are well established and agree with experimental results and their discussion. Experimental results are very interesting but the following points are to be considered for modification:
Firstly, Figure 8. are the Optical micrographs along the transversal section for each deformed step in the R2 sample. Please explain the reason for the difference between the surface zone and the central zone. Besides, why the recrystallization is more adequate at ε=0.80 than at ε=1.00 at the surface zone?
Secondly, the paper has presented experimental data of the EBSD results. Here I would like to encourage the authors to quantitatively describe the microstructure evolution by using the EBSD results such as DRX grain size, grain boundary misorientations, texture, etc., to reveal the detailed information related to thermomechanical processing.
Finally, The English of the paper needs polishing.
Author Response
Thank you very much for your comments. They were very useful to improve the understanding of the manuscript. Please find attached the new version of the manuscript with the corrections marked in red.
First comment response:
In the analysis of Figure 8 (line 285), we have modified the text where the optical micrographs between the central and surface zones of the sample are compared for different strains. In this description (section 3.4), we only compare the obtained microstructures and are suggested the restoration mechanisms that may be acting in each zone. The explanation of why a lamination process generates different effective deformations along the sample thickness was previously demonstrated by other authors [39], and we mention it after the EBSD analysis (Section 3.5) in the last summary paragraph.
Regarding the question in the first comment: Although in the micrographs of Figure 8a, the ε=0.80 appears to have a smaller grain size than the ε=1.00, the method does not fully reveal the high-angle misorientation boundaries. We have added a comment in line 288 to reflect the limitation of the optical microscopy technique. The conclusion that ε=0.80 is more adequate than ε=1.00 in grain refinement may be an artefact of the technique and/or the region chosen for the micrograph. In conclusion, we do not believe that there is a higher grain refinement at lower deformation.
Second comment response:
We had already considered many variants to show information regarding grain sizes and misorientations, either as disorientation between grains or as internal defects by Grain Axes Misorientations. However, we appreciate reviewer´s suggestion to include information about textures to enrich and complement the interpretation of the restoration mechanisms. The added information can be found in the paragraph of line 390 and in the new Figures 11 and 12. The new results presented here strengthen the conclusions reported in our previous manuscript.
Final comment response:
Typing corrections were made throughout the manuscript, along with other modifications mentioned by the other reviewers.

Author Response
Thank you very much for your comments. They were very useful to improve the understanding of the manuscript. Please find attached the new version of the manuscript with the corrections marked in red.
Comment response:
In reference [23] we have published how the BDTT is modified by the grain size, and although we mention it here to justify the framework of the work, our intention was not to revisit this topic. The thermomechanical process on the cylindrical ingot presented here is in fact the process we have used to produce the specimens of reference [23], as is mentioned in the introduction (line 74). However, we believe your suggestion is appropriate and we have added this to the last paragraph of the manuscript. We explain how the microstructure obtained here can be related to the BDTT and how it can be improved in the future (see line 438).
Additionally, we have expanded the information presented in the introduction regarding our previously explored link between the BDTT and microstructure (ref. [23]).
Response to the additional comment 1:
Since the characterisation of the carbides does not seem convincing to the reviewer, according to the results shown, and since it is not relevant to the aim of the present work, we have decided to remove the TEM carbides type characterisation. In the corrected manuscript version, we have kept the morphological and distribution description of the vanadium carbides observed by SEM, but we do not include the identification of the carbide types. We will leave this topic pending for a future publication, which will deal with the characterisation in more depth and with better statistics of the carbides formed in this alloy system.
Response to the additional comment 2:
The term “calm air cooling” was replaced by air cooling (natural convection). See lines 115, 259, and 263.
In turn, we have modelled the cooling rates of air-cooling for the specimens used. A section has been added to the Supplementary Materials, including the calculations' details and estimated cooling rates.
Response to the additional comment 3:
Figure numbering errors in the last paragraph have been corrected.

Round 2
Reviewer 3 Report
The authors addressed my concerns by removing the TEM data and not addressing the type of carbides in the alloy. This is a weakness but not the focus of the paper. Also, the authors add a bit more verbiage about the BDTT without providing any data instead pointing out that they predict that the nil ductility temperature can be reduced to a predicted 160°C and possibly lower. Again, not as convincing as actual data would be but not the focus of the paper. Finally, the authors did provide cooling rate predictions in the supplementary material as requested.